# Mung Bean (*Vigna radiata* (L.) R. Wilczek) from Burkina Faso Used as Antidiabetic, Antioxidant and Antimicrobial Agent

**DOI:** 10.3390/plants11243556

**Published:** 2022-12-16

**Authors:** Jeanne d’Arc Wendmintiri Kabré, Durand Dah-Nouvlessounon, Fatoumata Hama-Ba, Abiola Agonkoun, Felix Guinin, Haziz Sina, Arnaud N. Kohonou, Pascal Tchogou, Maximien Senou, Aly Savadogo, Lamine Baba-Moussa

**Affiliations:** 1Laboratory of Applied Biochemistry and Immunology, Department of Biochemistry and Microbiology, Joseph KI-ZERBO University, Ouagadougou 03 BP 7021, Burkina Faso; jeannedarc.kabre@yahoo.fr (J.d.W.K.); alysavadogo@gmail.com (A.S.); 2Laboratory of Biology and Molecular Typing in Microbiology, Department of Biochemistry and Cell Biology, Faculty of Science and Technology, University of Abomey-Calavi, Cotonou 05 BP 1604, Benin; dahdurand@gmail.com (D.D.-N.); agonkoun@gmail.com (A.A.); sina.haziz@gmail.com (H.S.); kohonouarnaud60@gmail.com (A.N.K.); 3Food Technology Department/Institute of Research in Applied Sciences and Technologies (IRSAT)/National Center for Scientific and Technical Research, Ouagadougou 03 BP 7047, Burkina Faso; hamafatou@gmail.com; 4Laboratory of Physiopathology/Molecular Pharmacology and Toxicology, Department of Animal Physiology, Faculty of Science and Technology, University of Abomey-Calavi, Cotonou 01 BP 4521, Benin; guinninf@gmail.com; 5Experimental and Clinical Biology Laboratory, National School of Applied Biosciences and Biotechnologies, National University of Science, Technology, Engineering and Mathematics (UNSTIM), Dassa-Zoumé 01BP 1471, Benin; tchopass2@gmail.com (P.T.); senouxim@yahoo.fr (M.S.)

**Keywords:** mung bean, bioactive molecule, biological activities, health food

## Abstract

Chronic non-communicable diseases are becoming more and more recurrent and require the addition of functional foods in our eating habits. Legumes due to their composition in biomolecules could meet this need. Much used in Chinese medicine, the mung bean arouses interest in Burkina Faso. The objective of this study is to perform phytochemical profiling and to evaluate certain biological properties of the mung bean in its natural or germinated state. Qualitative phytochemical screening was carried out by precipitation and differential staining tests. The antimicrobial activity was tested on in vitro growth by the agar medium diffusion method. DPPH and FRAP methods were used to assess antioxidant activity. The antidiabetic activity of hydroethanolic extracts was evaluated on rats rendered diabetic by streptozotocin, with metformin as a reference molecule. Phytochemistry has revealed the presence of phenolic compounds and derivatives in the mung bean, whether in its natural state (MBN) or in its germinated state (MBG). Only the MBG exhibits antimicrobial activity on 70% of the strains used. It appears that the MBG has a reducing power of the DPPH radical with an IC_50_ of 28 mg/mL compared to the same extract of the MBN, which had an IC_50_ of 32.5 mg/mL with a difference (*p* < 0.05) between the extracts. MBN extracts at a dose of 300 milligrams per kilogram of body weight (mg/kg.bw) showed a reduction (*p* < 0.0001) in glycaemia and kept the body weight of the animals constant throughout the treatment. In addition, the MBN regulated the level of total cholesterol, tryglicerides of LDL, ASAT, ALAT, urea and creatine. These results show that the mung bean grown in Burkina Faso is a health food, which, integrated into dietary habits, could contribute to the prevention of chronic diseases.

## 1. Introduction

Diabetes mellitus (DM) is a chronic carbohydrate metabolism disease characterized by hyperglycemia, and it is caused by a total lack of insulin, insufficient secretion or synthesis of insulin or peripheral resistance to the insulin effect [1]. DM affects a large number of people, especially in low- and middle-income countries. According to the World Health Organization (WHO), about 422 million people worldwide have DM, and 1.6 million deaths are directly linked to DM or DM-related complications [2]. Furthermore, its prevalence is expected to increase, reaching 578 million people by 2030 and 700 million people by 2045 [3]. Access to diagnosis and treatment is difficult for many patients with DM, which raises mortality and increases the likelihood of complications [4]. The by-products of cellular metabolism leading to the accumulation of free radicals are the cause of many diseases such as diabetes, cancer and obesity [5]. Despite the diversity of glucose-lowering drugs, diabetes and its complications remain a public health problem, hence the need to explore functional food approaches to solutions. Poor eating habits accompanied by a notoriously sedentary lifestyle are among the causes. In order to improve health status and prevent these diseases, a diversity of functional foods has been recommended by several global health organizations, prompting a call for serious changes in eating habits [6,7]. Among this diversity of functional foods, legumes rank high as important sources of macronutrients, micronutrients and biomolecules that can contribute to solving the problem of dietary and therapeutic deficiency in the world and particularly in Africa [7]. From the *Fabacae* family, the mung bean is one of the main legumes cultivated in India and much used in Chinese medicine due to its biomolecule composition according to [8,9]. Mung bean seeds and sprouts are consumed worldwide as excellent functional foods that reduce the risk of many diseases [10]. As a food, mung beans contain balanced nutrients, including protein and dietary fiber, and significant amounts of bioactive phytochemicals [11]. High levels of proteins, amino acids, oligosaccharides and polyphenols in mung beans are thought to be the main contributors to the antioxidant, antimicrobial, anti-inflammatory and antitumor activities of this food and are involved in the regulation of lipid metabolism and contribute to its pharmacological activities [12]. The cotyledons and seed coats of the mung bean are rich in phenolic acids, flavonoids and tannins according to some authors [13,14]. The mung bean has been subjected to phytochemical analysis, which identified the presence of tannins, saponins, phenols, alkaloids, terpenoids, steroids, avonoids and glycosides. Per gram of *V. radiata* seed extract, the total phenolic content was 43.12 ± 3.14 mg gallic acid equivalents, while the total flavonoid content was 38.35 ± 2.6 mg quercetin equivalents [15]. In the diabetes treatment, [15] reported that the levels of glycated hemoglobin (HbA1c) and blood sugar were significantly reduced by the *Vigna radiata* extract from Ethiopia. Newly introduced and promoted in Burkina Faso, the mung bean is reportedly resilient to the effects of climate change. With a high nutritional value, it is cultivated and consumed in a variety of dishes [16]. The mung bean has several varieties, about 2000 [10,11]. Many authors have reported that agronomic, soil and meteorological factors impact on the amount of iron in the soil and in turn influence the iron concentration in mung bean seeds [12,13]. It was therefore necessary to verify the biochemical and nutritional composition of mung beans grown in Burkina Faso and to assess their biological activities. Authors have also reported the good anti-inflammatory and anticancer activity of the mung bean and its health benefits [14,15,16,17,18]. The objective of this study was to explore free radical scavenging activities, antimicrobial activities and antihyperglycemic capacities, preceded by qualitative profiling of chemical constituents present in mung beans grown in Burkina Faso.

## 2. Results

### 2.1. Phytochemical Screening

The results of the phytochemical screening carried out on MBG and MBN seed powder are presented in Table 1. It can be seen that the presence of secondary metabolites varies according to the samples. Indeed, the MBN sample contains 66.66% of the secondary metabolites sought while the MBG sample contains 50%. Secondary metabolites such as flavonoids, anthocyanins, saponosides, mucilage and terpenoids are common to both samples while catechic and gallic tannins are absent only in the MBG sample.

### 2.2. Antimicrobial Activity

#### 2.2.1. Sensitivity Testing of Microbial Strains (Antibiogram)

The antibiogram performed on the different extracts revealed that all (100%) of the microbial strains were resistant to all MBN extracts. In contrast, the MBG extracts were active on 54.54% of the tested strains. The diameters of inhibition varied according to the type of extract (Figure 1). The largest diameter obtained with ethanolic extract was 26.75 ± 0.35 mm (*Candida albicans*), that obtained with methanolic extract was 25.4 ± 0.56 mm (*Staphylococcus epidermidis*) and 22.75 ± 0.35 mm (*Staphylococcus aureus*) with hydroethanolic extract. The analysis of the data shows that both ethanolic and methanolic extracts are active over time with an increase in the inhibition diameter after 48 h. This observation shows the strong action of these extracts on the sensitivity of the tested strains. The comparative effect of the extracts shows a variation (*p* < 0.0001) between the inhibition capacity of the extracts towards the tested microbial strains (Figure 1).

The antibiograms of the different MBG extracts after 24 h of incubation are presented in the Figure 2 below. The inhibition diameters vary from one extract to another and from one microorganism to another.

#### 2.2.2. Minimum Inhibitory Concentrations (MICs) and Minimum Bactericidal Concentrations (MBCs)

The MICs and MBCs obtained with the different MBG extracts on susceptible microbial strains vary from one extract to another (Table 2). Indeed, the lowest MIC obtained with the ethanolic extract was 3.12 mg/mL (*Staphylococcus epidermidis*), while that obtained with the hydroethanolic extract was 6.25 mg/mL (*Pseudomonas aeruginosa*) and that obtained with the methanolic extract was 3.12 mg/mL (*Escherichia coli*). However, among the three extracts, the lowest MBC, 6.25 mg/mL, was obtained with the ethanolic extract on the *Staphylococcus epidermidis* strain. The ratio of the two parameters shows that all MBG extracts showed bactericidal effects on 66.67% of susceptible strains. Indeed, all the active MBG extracts showed a bacteriostatic effect on *Pseudomonas aeruginosa* and *Escherichia coli*. Taking into account the degree of sensitivity of the strains, we notice that Staphylococcu epidermidis is the most sensitive strain, followed by *Candida albicans*, *Staphylococcus aureus* and *Escherichia coli* O157 (Table 2).

### 2.3. Antioxidant Activity of Extracts

The antioxidant power of MBN and MBG extracts was summarized in Table 3. Using the DPPH method, it was noticed that the inhibition percentages vary from 54.26 ± 0.06% (MBG ethanolic extract) to 88.32 ± 0.06% (MBG water–ethanol extract). In addition, the lowest IC_50_, which reflects the highest DPPH radical reduction power, was obtained with the hydroethanol extract of MBN (IC_50_ = 28.65 ± 0.21 mg/mL) with an antioxidant activity index (AAI = 1.74 ± 0.01) lower than that obtained with the reference molecule BHA (AAI = 23.43 ± 5.08). The comparative effect of the active extracts shows that the hydroethanolic extract of MBN showed the highest DPPH-reducing power while the ethanolic extract of MBG exhibited the lowest DPPH-reducing power with an IC_50_ = 80.00 ± 4.24 mg/mL and an antioxidant activity index AAI = 0.62 ± 0.03 (Table 3). With the FRAP method, MBN and MBG extracts confirmed their variable antioxidant activities. The inhibition percentages varied from 77.51 ± 0.71% (MBN ethanolic extract) to 83.40 ± 0.13% (MBN methanolic extract). In addition, the hydroethanolic extract of MBG showed the highest chelating power of ferric ion with the lowest IC_50_ of 11.06 ± 0.40 mg/mL. Compared to the reference molecule used, the activity of this extract is 5.58 times smaller than that of quercetin (IC_50_ = 1.98 ± 0.18 mg/mL).

### 2.4. Antidiabetic Activities of Extracts

#### 2.4.1. Changes in Blood Glucose and Body Weight

In order to test the antidiabetic activity of the ethanolic extract of the MBN and MBG, female wistar rats were made diabetic with STZ. Intraperitoneal injection of STZ effectively rendered the rats diabetic with blood glucose levels ranging from 251 mg/dL to 489 mg/dL, or a 92.6% induction rate. The two-factor ANOVA analysis of variance showed a difference in blood glucose levels between the different groups (*p* < 0.0001). This variation was also observed at the level of induction and treatment days (*p* < 0.0001) as well as with the interaction between groups and treatment days (*p* < 0.001). Repeat, however, had no variation (*p* > 0.05) on blood glucose and body weight.

During induction, blood glucose levels changed with time (*p* < 0.0001). Indeed, they increased from 102.81 ± 9.01 mg/dL (Day 0) to 419.64 ± 55.18 mg/dL (Day 3) and to 427.31 ± 53.7 mg/dL (Day 7). However, the influence of induction on body weight showed no significant difference (*p* > 0.05) on the evolution of the body weight of the animals (Table 4). Nevertheless, the weight changed from 201.75 ± 13.44 g (Day 0) to 199.62 ± 13.08 g (Day 7).

#### 2.4.2. Antihyperglycemic Effect of Mung Bean Extracts

##### Evolution of Blood Glucose during Treatment

The effect of the extracts (MBN and MBG) and metformin (reference molecule) on the evolution of the blood glucose and body weight of the rats is presented in Table 4. The evolution of the glucose concentration of the negative control group (G1) shows that feeding did not influence (*p* > 0.05) the blood glucose concentration during the 14 days of the experiment (Table 4). Glucose concentration remained almost stable (113.5 ± 2.59 mg/dL at D0 to 104 ± 1.15 mg/dL at D14) throughout the experiments. Concerning the rats of the positive control group (G2), the evolution of blood glucose levels shows that the rats were maintained diabetic throughout the 14 days of experimentation. The different treatments show the effectiveness of the extracts (MBN and MBG) and of metformin (reference molecule) in reducing the blood glucose level. A variation in the sugar level between the seven groups of treated rats was noticed. Indeed, the comparative effect of the extracts shows a variation in the effectiveness according to the type of extract and the concentration. Regardless of the concentration used, the MBN extract showed the strongest blood-sugar-lowering power. Indeed, the MBN extract at 300 mg/kg.bw (G6) significantly (*p* = 0.001) lowered blood glucose from 434.33 ± 22.67 mg/dL (D0) to 167.33 ± 39.71 mg/dL (D7) and then to 108 ± 2.64 mg/dL (D14). The efficacy of this extract is followed by that of the MBN extract at 150 mg/kg.bw. Consequently, the MBN extract at 150 and 300 mg/kg.bw was more active than the metformin used as a reference molecule. Metformin reduced (*p* < 0.001) blood glucose levels from 425 ± 24.82 mg/dL (D0) to 267 ± 19.05 mg/dL (D7) and then to 166.5 ± 20.49 mg/dL (D14). Moreover, the MBG extract at 300 mg/kg.bw was more active than the MBG extract at 150 mg/kg.bw, showing a dose–response activity.

##### Effect of Extracts on Weight Evolution

The action of the extracts (MBG and MBN) and metformin (reference molecule) on the variation in body weight is presented in Table 4. The analysis of the data presented in Table 4 shows through the evolution of the weight load of the negative control group (G1) that the rats had a good food ration. The weights of the rats in this group evolved progressively (*p* < 0.001) from 199 ± 8.66 g (D0) to 201 ± 8.08 g (D7) and 204 ± 8.66 g (D14). In contrast to the observations made in the negative control group, the positive control rats (G2) gradually (*p* < 0.0001) lost their weights during the experiments, showing that the induction of diabetes effectively sickened the positive controls (G2). The weight load decreased from 200 ± 4.61 g (D0) to 199 ± 4.61 g (D7) and to 196.5 ± 4.90 g (D14). Furthermore, although MBG and MBN extracts at 150 mg/Kg.bw as well as 300 mg/kg.bw did not allow the rats to significantly recover their body weights after 14 days of treatment, these extracts (MBG and MBN) did however allow the rats to lose no more weight depending on the groups and days of treatment compared to the positive control (Table 4). Indeed, at 300 mg/kg.bw (G4) for the MBG extract, the weight load increased from 199.33 ± 14.84 g (D0) to 199.66 ± 14.74 g (D7) and 203.33 ± 14.51 g (D14). For MBN extracts, at a concentration of 150 mg/kg.bw (G5), the weight load remained unchanged after 7 days of treatment (200.33 ± 6.35 g at the 1st and 201.66 ± 6.11 g at the 7th day) and increased to 205.33 ± 6.04 g at the 14th day. This was not the case for the same MBN extract at a concentration of 300 mg/kg.bw (G6), which showed a weight load gain ranging from 199.66 ± 6.88 g (D0) to 204 ± 4.50 g (D7) and to 207.33 ± 5.69 g (D14). A similar evolution (*p* < 0.001) was also observed for metformin (199.5 ± 8.37 g (D0) to 202 ± 8.66 g (D7) and to 205.5 ± 8.37 g (D14)). All these observations show that the MBN extract showed better efficacy through the maintenance and evolution of body weight despite the presence of STZ-induced disease.

##### Biochemical Parameters

Blood function: Blood Count (CBC)

The action of the extracts (MBG and MBN) and metformin (reference molecule) on the variation in erythrocytometry and leukocytometry is presented in Table 5. Compared to the negative control (G1) of normal rats without diabetes and without treatment, the diabetic animals in group G2 have lower hemoglobin (17.15 ± 0.08 vs. 19.00 ± 0.00), red blood cell (5.77 ± 0.03 vs. 6.38 ± 0.00) and hematocrit (52.5 ± 0.28% vs. 58.00 ± 0.00%) content. There is also a high content of white blood cells (7.25 ± 0.95 g/dL vs. 5.8 ± 0.76 g/dL), with 24.00 ± 5.65% of lymphocytes and 76.00 ± 5.65% of neutrophils. From these values, we can see that the induction of diabetes caused anemia in diabetic animals through a decrease in red blood cells and an increase in white blood cells compared to the negative control. In contrast, treatments with the MBG (at 150 and 300 mg/kg bw), MBN at 150 mg/kg bw and metformin showed an improvement in red blood cell values in the sense of a correction of anemia, and a decrease in white blood cell content. However, animals treated with the MBN 300 mg/kg bw (G6) showed lower red blood cell content (5.33 ± 0.38 g/dL) and 15.86 ± 1.23 g/dL hemoglobin with 48.66 ± 3.71% hematocrit. This suggests the presence of anemia after 14 days of treatment. These same animals had a white blood cell count similar to that of the diabetic control group (G2) with 7.44 ± 1.12 g/dL vs. 7.25 ± 0.95 g/dL.

##### Lipid Profile

The action of the extracts (MBG and MBN) and metformin (reference molecule) on the variation in the lipid profile is presented in Table 6. The one-factor ANOVA analysis of variance shows an overall variation (*p* < 0.001) between the lipid parameters. Compared with the negative control group (G1) that received no treatment, the Student–Newman–Keuls (SNK) test shows that individuals in the positive control group (G2) with diabetes showed a higher total cholesterol level (4.37 ± 0.05 mmol/L) while the rats in the non-diabetic negative control group (G1) and those in the G6 group (MBN 300 mg/kg.bw) showed the lowest total cholesterol levels of 3.41 ± 0.05 mmol/L and 3.36 ± 0.02 mmol/L, respectively. The same observation was made for total triglycerides where the G2 group showed the highest level (4.80 ± 0.01 mmol/L) while the treatment with MBN extracts at 300 mg/kg.bw (G6) reduced this level to 3.58 ± 0.01 mmol/L. The same trend was observed with LDL content. Indeed, diabetic rats G2 showed the highest LDL level (0.51 ± 0.01 mmol/L) while non-diabetic rats (G1) and those treated with MBN 300 mg/kg.bw showed the lowest levels, 0.20 ± 0.01 mmol/L and 0.23 ± 0.01 mmol/L, respectively. These different observations show that the induction of diabetes in rats increased the lipid profile of diabetic rats that were fed the same diet as the non-diabetic control rats. Furthermore, after 14 days, all the different treatments performed with MBN, MBG and metformin extracts reduced the lipid profile of diabetic rats. Indeed, we also observe an activity according to each extract on the one hand and their concentrations on the other hand. At 300 mg/kg.bw, the two extracts (MBN and MBG) showed a strong activity by lowering the total cholesterol level to 23.12% compared to the positive control, the total triglycerides to 25.42% and the LDL to 54.91%. Of the two extracts, the MBN extract at 300 mg/kg.bw showed the highest activity by reducing total cholesterol from 4.37 ± 0.05 mmol/L to 3.36 ± 0.02 mmol/L, total triglycerides from 4.80 ± 0.01 mmol/L to 3.58 ± 0.01 mmol/L and LDL from 0.51 ± 0.01 mmol/L to 0.23 ± 0.01 mmol/L. It should also be noted that the MBN extract at 300 mg/kg.bw showed a stronger activity than metformin, which also showed good activity in reducing total cholesterol (3.52 ± 0.01 mmol/L), total triglyceride (3.72 ± 0.04 mmol/L) and LDL (0.25 ± 0.00 mmol/L).

##### Liver Function

The action of the extracts (MBG and MBN) and metformin (reference molecule) on liver function was evaluated by measuring the enzymatic activity of transaminases, alanine amino transferase (ALAT) and aspartate amino transferase (ASAT). Enzyme activity is the amount of enzyme present in a tissue or organ. It is difficult to measure the amount of enzyme in units of mass or molar concentration; enzyme activity is defined in terms of reaction rate. It is the equivalence of a maximum rate and is expressed in International Units (I.U.). The analysis of variance ANOVA of the results obtained show that the induction of diabetes resulted in a variation (*p* = 0.001) in the amount of enzyme (ASAT) between the positive and negative control (Table 7). Indeed, the ASAT/GOT level was 123 ± 0.00 IU/L for the diabetic group (G2) vs. 109 ± 4.04 IU/L for the non-diabetic group (G1). The Student–Newman–Keuls (SNK) test shows that the enzymatic activity is not significant between these two groups. Nevertheless, the different treatments performed show that the MBG extract at 150 mg/kg.bw reduced the amount of ASAT/GOT of the rats made diabetic from 123 ± 0.00 I.U/L to 96.66 ± 4.33 I.U/L. When the concentration of the same extract (MBG) was increased to 300 mg/kg.bw, the amount of ASAT/GOT was reduced to 70.33 ± 12.91 I.U/L, showing that increasing the concentration of the extract reduced the amount of enzyme by 57.17%, which could reflect over a long period of time a dysfunction of the liver function. The same observation was made with ALT/GPT, which went from 91 ± 5.19 I.U/L to 88.33 ± 6.93 I.U/L (MBG at 150 mg/kg.bw) and then to 64.33 ± 13.48 I.U/L (MBG at 300 mg/kg.bw). For the MBN extract, the same trend was observed and the MBN extract at 300 mg/kg.bw reduced the amount of ASAT/GOT from 123 ± 0.00 I.U/L to 72.66 ± 9.52 I.U/L and ALAT/GPT from 91 ± 5.19 I.U/L to 67.66 ± 10.72 I.U/L. The comparative effect of the extracts shows that the MBG extract exhibited the best activity. This activity was also higher than that of metformin, which reduced ASAT/GOT from 123 ± 0.00 I.U/L to 99.00 ± 1.73 I.U/L (G7) and ALAT/GPT from 91 ± 5.19 I.U/L to 85.00 ± 2.30 I.U/L (G7).

##### Renal Function

The effect of extracts (MBG and MBN) and metformin (reference molecule) on the variation in urea and creatine content is presented in Table 7. The overall analysis of variance of the data shows that the induction of diabetes influenced renal function by variation (*p* < 0.05) in creatine content. Indeed, the comparison between the two controls shows that the urea content was 0.25 ± 0.06 g/L in the non-diabetic group (G1) vs. 0.52 ± 0.00 g/L in the diabetic group (G2). Similarly, creatine content was 8.4 ± 1.84 mg/L in the non-diabetic group (G1) vs. 16.55 ± 0.20 mg/L in the diabetic group (G2). Thus, the induction of diabetes increased urea and creatine levels by 51.93% and 49.25%, respectively. After 14 days of treatment, the highest activity was observed with the MBG extract at 300 mg/kg.bw (G6), which reduced the urea content from 0.52 ± 0.00 g/L to 0.29 ± 0.03 g/L. The same extract (MBG at 300 mg/kg.bw) reduced creatine content from 16.55 ± 0.20 mg/L to 9.2 ± 0.90 mg/L. It should also be noted that the activity of the MBG extract at 300 mg/kg.bw (G6) is higher than that of metformin (G7), which reduces urea content to 0.36 ± 0.02 g/L and creatine content to 11.85 ± 1.01 mg/L.

##### Pearson Linear Correlation

In order to better understand the relationship between the different variables related to biochemical parameters, a standard Pearson linear correlation was performed (Table 8). A high correlation was observed between the variables evaluated. Indeed, the total cholesterol level is significantly (*p* < 0.001) positively correlated with the triglyceride and LDL levels, which confirms that these three parameters are closely related. However, a positive and significant correlation was also observed between total cholesterol (TChol) and urea (*p* < 0.01); triglycerides (TryG) and urea (*p* < 0.05); and LDL and urea (*p* < 0.01), which shows that these three lipid parameters could have an influence on urea secretion. In contrast, these three lipid parameters had no significant correlation with blood function (HB, HT, RBC and WBC). However, considering the liver function, we observe a positive and significant correlation (*p* < 0.05) between LDL and ASAT, which shows that the LDL level influenced the alanine amino tranferase (ASAT) activity. Similarly, LDL was positively correlated with creatine (*p* < 0.01); this shows that apart from blood parameters, LDL influenced liver and kidney functions. Regarding the blood count (CBC) parameters, there was a positive and highly significant correlation between HB and HT (*p* < 0.001); HB and NR (*p* < 0.001); and HT and RBC (*p* < 0.001). It should be noted that most blood parameters are negatively correlated with liver and kidney functions (Table 8).

#### 2.4.3. Histopathology

A safety evaluation of the extracts in diabetes shows that in the subchronic oral toxicity test of the MBG extract at 150 mg/kg.bw (Figure 3B) or MBN extract at 150 mg/kg.bw (Figure 3C), hepatocytes (H) in the centrolobular regions showed no visible atypia and were like those in the non-diabetic control (Figure 3A). Hepatocytes were well arranged in a cord around the centrilobular veins (CV). There was good visibility between the hepatocytes and the sinusoids (S). Around the portal spaces (PS), the liver parenchyma was also typical with the MBG 300 mg/kg.bw extract (Figure 3E) or MBN 300 mg/kg.bw (Figure 3F), as with metformin (Figure 3D), indicating an absence of liver toxicity.

## 3. Discussion

Several groups of secondary metabolites are present in mung beans grown in Burkina Faso according to our results. A variety of metabolites are present in both natural and germinated mung bean seeds and are summarized in Table 1. The cotyledons and seed coats of mung beans are rich in phenolic acids, flavonoids and tannins according to [13,14]. Moreover, [19,20] reported that the contents of free phenolic acid and bound phenolic acid in mung bean varieties ranged from 16.68 to 255.51 μg/g and 2284.53 to 5363.75 μg/g, respectively. Some biological activities such as antimicrobial, antioxidant and antidiabetic activity are promoted by biomolecules such as tannins, flavonoids and polyphenols found in plain and sprouted mung beans. Thus, the biological activities attributed to the mung bean depend on its composition in secondary metabolites reported by several authors such as [21,22]. In [23], it was reported that polysaccharides from mung bean seed coats could improve gut health. MBG extracts showed good antimicrobial activity on 70% of the tested strains in contrast to MBN extracts, as shown in Figure 1. Several authors also reported that germinated mung beans had more phenolic compounds than raw seeds [14,24,25,26] and also reported high antimicrobial activity with mung bean germ extracts against *Helicobacter pylori*. MBGs contains phenolic compounds that have been reported by several authors to reduce the risk of cancer and have antibacterial and antiviral properties [10,18]. The most plausible hypothesis would be the presence of intermediate metabolites in MBGs, promoted by the process of germination. Germination, in addition to causing physical changes in the seed, contributes to changes in biochemistry; [27] showed that soluble dietary fiber content increases 3–4 fold following germination. These intermediate metabolites present in MBG have an inhibitory effect on the one hand and a bactericidal effect on the other hand for the tested strains with a significant difference between the inhibition diameters of the different strains, which vary from one extraction solvent to another with *p* < 0.05. The bactericidal effect was observed on both *Gram +* and *Gram*—bacteria as well as on fungal strains. Our results are in agreement with those of Shaoyun [28], who reported that, for the first time, a plant lysozyme, speaking of the mung bean, exerted antifungal action against *Fusarium oxysporum, Fusarium solani, Pythium aphanidermatum, Sclerotium rolfsii* and *Botrytis cinerea*, but also on *Staphylococcus aureus* Gram + bacteria. In [29], it was concluded from their work that sprouted mung bean extracts have potent antiviral activities attributable to the potential of the sprouted mung bean to induce antiviral cytokines.

The results show a significant difference (*p* = 0.0005) between the reducing powers of the different extracts of both MBN and MBG. The MBN and MBG extracts show antiradical properties with a highly significant difference (*p* = 0.0001) between the different samples due to the ability of the solvents to extract the biomolecules in the different samples. The hydroethanol extract of MBG had a high reducing power of DPPH radical (88.37 ± 0.06%) with an IC_50_ of 28 mg/mL compared to the same extract of MBN in which it was 85.75 ± 0.06 with 32.5 mg/mL as IC_50_. Several authors have reported the antioxidant activity of mung beans, which differs from one variety to another. In [8,20], it was reported that sprouted mung bean and plain mung bean seeds had an IC_50_ of 4.148 and 4.93 mg/mL, respectively. The evaluation of reducing power by FRAP method showed that the hydro-ethanolic extracts of both MBN and MBG, with 34 mg/mL and 11 mg/mL IC50, respectively, had more reducing power than the ethanolic or methanolic extracts of MBN and MBG. The results of [30] showed the good anti-inflammatory and anticancer activities of the hydroethanol extracts of mung beans. These antioxidant properties of plain and sprouted mung beans could have a beneficial effect on degenerative diseases, such as diabetes, that set in with oxidative stress.

Rats made diabetic by STZ induction showed severe hyperglycemia related to a probable decrease in endogenous insulin secretion and release. Rats treated with MBN and MBG extracts showed a significant decrease in blood glucose level. These results indicate that mung beans grown in Burkina Faso, whether plain or sprouted, produce antihyperglycemic activity and this hypoglycemic activity can be attributed to insulin sensitivity. The results showed that the ethanolic extract of mung bean seed coat significantly improved insulin sensitivity [12]. Moreover, STZ induces selective destruction of pancreatic β cells leading to impaired glucose utilization resulting in hyperglycemia, but leaving many surviving β cells, which can be regenerated according to [31]. The increase in insulin could be related to the regeneration of pancreatic β cells destroyed by STZ according to the work of [32]. In addition, this regulation of blood sugar could also be explained by the presence of polyphenols and flavonoids in mung beans having the ability to regulate the gut microbiota, thus protecting the host against pathogen invasion and the risk of type 2 diabetes according to the work of [33].

Our results also show that oral administration of MBN and MBG extracts decreases triglycerides, total cholesterol and LDL, indicators of chronic disease development. The decrease in serum triglycerides may be associated with changes in total serum magnesium concentration. The the methanol extract of *Vigna radiata* seeds also moderated lipid profiles, AST, ALT and glycated hemoglobin while restoring liver glycogen and insulin levels in diabetic mice, suggesting its potential advantages in reducing some of the consequences of diabetes [15]. There is increasing evidence of a role for magnesium in modulating serum lipids and lipid absorption in macrophages [34]. Some flavonoids in plain and sprouted mung beans have hypoglycemic properties as they improve the altered glycemic and oxidative metabolisms of diabetic states. These biomolecules also exert a stimulating effect on insulin secretion by altering the Ca++ concentration according to [35]. Previous studies have shown that mung bean consumption is associated with the modulation of lipid metabolism. Mung bean protein was found to dose-dependently reduce plasma lipid levels such as total cholesterol, triglycerides and low-density lipoprotein cholesterol, according to [20,36], who concluded that the mung bean could be a nutraceutical food to maintain glycemic control and mitigate diabetes complications.

The biochemical results show a dysfunction in the hepatic function of diabetic rats. Moreover, the different treatments carried out show that MBN and MBG extracts as well as metformin have a beneficial effect on the regulation of muscle creatine and could attenuate the occurrence of renal failure in diabetic rats. The anemia caused by the induction of diabetes in rats, and materialized by decreases in hemoglobin, hematocrit and red blood cells, is regularized with the gavage with MBG extracts with a significant difference in comparison with the rats of the positive control group. Whole or sprouted mung beans were found to be effective hepatoprotective agents capable of decreasing liver enzyme activities and liver histopathology in a dose-dependent manner according to the work of [37]. The significant increase in the number of neutrophils and white blood cells in diabetic rats is indicative of an infection caused by the effects of STZ. In addition, MBN and MBG extracts as well as metformin resulted in an increase in body weight in diabetic rats.

## 4. Materials and Methods

### 4.1. Sample Preparation

The plant material used consisted of mung bean (*Vigna radiata*) seeds collected from the Belwet structure located in district 2 of the Ouagadougou municipality, the main promoter of mung beans in Burkina Faso. The mung bean was used in two forms as raw material. These are the natural form (MBN) and the sprouted form (MBG). The natural mung bean sample (MBN) was obtained by washing the natural mung bean seeds with demineralized water and then drying them at 60 °C for 12 h. The sprouted mung bean sample (MBG) was obtained by hydrating the natural mung bean seeds with deionized water for 12 h before germinating them between two bedded fabrics for 48 h. After drying at 60 °C for 12 h, the rootlets were removed. A stainless-steel grinder (IKA Bro-03-PATACE-018) was used to grind the different samples.

### 4.2. Phytochemical Screening

The phytochemical screening performed is a qualitative chemical analysis based on staining or precipitation reactions. It was performed directly on the MBN and MBG powder. The method used is that of [38], adapted to the conditions of the Laboratory of Biology and Molecular Typing in Microbiology of the University of Abomey Calavi in Benin.

### 4.3. Preparation of Extracts

The different samples underwent three types of extraction each: ethanolic, hydroethanolic and methanolic extraction. The extraction method described by [39] was used. For this purpose, a mass of 50 g of powder of the different samples of MBN and MBG was macerated in 500 mL of ethanol, methanol and 30% ethanol for ethanolic, methanolic and hydroethanolic extracts, respectively, and then left under stirring for 72 h. The macerates were then filtered 3 times with absorbent cotton and once with Whatman paper. The resulting concentrates were then dried at 40 °C in an oven. The powders thus collected after drying constitute our extracts for the study of different biological activities.

### 4.4. Evaluation of the Antimicrobial Activity of Extracts

The microorganisms used include bacteria and a yeast. They are ten reference strains and one strain isolated and characterized by [40]. The strains are composed of the following:

Five Gram + bacteria (Staphylococcus aureus ATCC 29213, Staphylococcus epidermidis T22695, Micrococcus luteus ATCC 10240, Streptococcus oralis NCTC 8029 and Enterococcus foecalis ATCC 29212).

Four Gram—bacteria (Pseudomonas aeruginosa ATCC 27853, Proteus miriabilis A24974, Proteus vulgaris A25015, Escherichia coli ATCC 25922).

A fungal strain (*Candida albicans* MHMR).

A foodborne strain of the genus staphylococcus (*Staphylococcus aureus*) isolated from meat products by [40].

#### 4.4.1. Preparation of the Preculture and the Antibiogram

The preculture was prepared from the 18–24 h old juvenile strains. An isolated colony of the test strain was taken and homogenized in 1 mL of nutrient broth before being incubated for 18–24 h at 37 °C. From this preculture, the inoculum with a microbial load of approximately 106 CFU/mL was obtained by dilution. The antibiogram was performed according to the disc method described by [41]. A total of 1 mL of the second decimal dilution of the 18–24 h preculture was used to inoculate a Petri dish containing a Mueller–Hinton (MH) culture medium by flooding. After inoculation, sterile 5 mm diameter Whatman No. 1 paper discs were placed on the dish using sterile forceps. Amounts of 30 µL of each of our extracts at 200 mg/mL concentration previously prepared in 30% DMSO were placed on the discs. The plates containing the impregnated discs were then left for 15 to 30 min at room temperature (25 °C ± 2 °C) for prediffusion of the substances before being incubated at 37 °C in the oven. The diameters of the possible inhibition zones were measured with a graduated ruler after an incubation time of 24 h and 48 h. For each extract, the experiment was performed in duplicate.

#### 4.4.2. Determination of the Minimum Inhibitory Concentration (MIC) and the Minimum Bactericidal Concentration (MBC)

The MIC is the lowest concentration of the compound at which the tested microorganism does not exhibit turbidity (no visible growth). The macro dilution method with visual observation described by [39] was used. After identifying the MIC, all tubes starting from the MIC to the high concentrations were plated with a platinum loop onto petri dishes containing MH agar medium. These plates were incubated in an oven at 37 °C for 24 h. Upon observation, the concentration of the extract that shows no microbial growth represents the MBC.

### 4.5. Evaluation of the Antioxidant Activity of Different Extracts

Two methods were necessary to conclude on the antioxidant activity of our extracts. The method described by [42] using DPPH (2,2-diphenyl-1-picryl-hydrazyl) as a relatively stable free radical, and the FRAP (ferric reducing antioxidant power) method following the protocol described by [43]. For the DPPH method, a 100 µg/mL methanolic solution of DPPH is mixed with each extract prepared from a 200 µg/mL stock solution. An amount of 1 mL of the DPPH methanolic solution was added to 1 mL of each extract, mixed, shaken and then incubated at room temperature and in the dark for 30 min. The absorbance was read at 517 nm against a blank consisting of DPPH and methanol. Butylated hydroxyanisole (BHA) was used as a reference molecule.

$$\%\;\mathrm{Inhibition}=\frac{\left(\mathrm{Absorbance}\;\mathrm{Blanc}-\mathrm{Absorbance}\;\mathrm{Extrait}\right)}{\mathrm{Absorbance}\;\mathrm{blanc}}\times 100$$


The concentration that inhibits 50% of DPPH (IC_50_) was determined graphically by the probit method. The antioxidant activity index (AAI) was calculated according to the formula proposed by Scherer and [44].

The evaluation of reducing power by FRAP method was performed by mixing 0.5 mL of the sample at different concentrations with 1 mL of phosphate buffer (0.2 M; pH = 6.6) and 1 mL of 1% potassium hexacyanoferrate [K3Fe (CN)6]. After incubating the mixture at 50 °C for 30 min, 1 mL of 10% trichloroacetic acid was added to stop the reaction, then the tubes were centrifuged at 3000 rpm for 10 min. An amount of 1 mL of the supernatant from each tube was mixed with 0.2 mL of 0.1% FeCl3 solution and allowed to stand in the dark for 30 min before measuring optical densities at 700 nm. Quercetin was used as a reference molecule. The antioxidant activity related to the reducing power of the extracts is expressed as reducing power (RP) using the following formula:
(1)
$$\mathrm{PR}=\frac{\left(\mathrm{Absorbance}\;\mathrm{Extrait}-\mathrm{Absorbance}\;\mathrm{Blanc}\right)}{\mathrm{Absorbance}\;\mathrm{Extrait}}\times 100$$


### 4.6. In Vivo Antihyperglycemic Activity of Extracts

The animal material used was Wistar rats with a health status free of specific pathogens, approximately eight weeks old and weighing 150 to 200 g. After two weeks of acclimatization at a constant temperature of 22 ± 2 °C under a 12/12 h light/dark cycle, the rats were divided into batches for the different tests. The animals were housed in integrated polypropylene cages with water pots and under hygienic conditions with standard rat food and free access to water. The body weight of the rats was recorded at the beginning and at the end of the experiment.

All handling procedures also took into account the guidelines validated by [45] for animal models of diabetes studies. The experimental protocol was approved by the Scientific Ethics Committee of the Doctoral School (Life Sciences) of the Faculty of Science and Technology (FAST) of the University of Abomey Calavi (UAC) under the number UAC/FAST/EDSV/1357006.

#### 4.6.1. Induction of Diabetes by Streptozotocin

Diabetes was induced by intraperitoneal injection of streptozotocin (STZ) (Sigma, St. Louis, MO, USA) freshly dissolved in 0.1 M of sodium citrate buffer pH 4.5 at a dose of 55 mg/kg body weight in female Wistar rats deprived of food for 16 h with free access to water. Streptozotocin is capable of inducing fatal hypoglycemia resulting from massive pancreatic secretion of insulin [46]. Animals were allowed to drink 5% glucose water overnight to overcome drug-induced hypoglycemia. After 7 days of streptozotocin (STZ) administration, fasting blood glucose was measured using a glucometer [47]. Fasting animals with blood glucose levels above 200 mg/dL were considered diabetic and selected for antidiabetic activity [48].

#### 4.6.2. Treatment of Diabetic Rats

Diabetic rats with blood glucose levels above 200 mg/dL were divided into six batches of 4 rats according to their body weights. A batch of 4 non-diabetic rats was also formed as a negative control batch. The animals were monitored with the hydroethanol extracts of MBN and MBG and metformin used as the reference antidiabetic.

Group 1: negative control group, receiving only sodium citrate buffer;

Group 2: untreated diabetic control group;

Group 3: diabetic group treated with MBG extract at a dose of 150 mg/kg bw;

Group 4: diabetic group treated with MBG extract at a dose of 300 mg/kg bw;

Group 5: diabetic group treated with MBN extract at a dose of 150 mg/kg bw;

Group 6: diabetic group treated with MBN extract at a dose of 300 mg/kg bw;

Group 7: diabetic group treated with metformin at a dose of 100 mg/kg bw;

The administration of the extracts and metformin was conducted orally through a feeding tube, and was conducted every morning during the 14 days that the experiment lasted.

#### 4.6.3. Determination of Biochemical Parameters

The blood glucose and body weight of the control and treated rats were monitored periodically throughout the experiment. The biochemical parameters were determined from blood samples taken from the rats of the different experimental groups. Blood collection was performed according to the experimental protocol employed by [49]. The animals were deprived of food 10 h before the collection performed from a retro-orbital sinus puncture without anesthesia. Some parameters were sought on whole blood while others were assayed in plasma obtained after centrifugation at 3000 rpm. Biochemical parameters, such as the following, were determined using specific kits, according to manufacturer’s instructions: hematocrit, hemoglobin, red and white blood cell count, leukocytes, neutrophils, blood glucose, alanine amino transferase and aspartate amino transferase (ASAT, ALAT) transaminases, total cholesterol, low-density lipoprotein, total triglycerides, creatinine and urea.

### 4.7. Statistical Analysis

Microsoft Excel 2016 spreadsheet software was used for data processing. Statistical Analysis System version 9.2 (SAS v. 9.2) software was then used for statistical analyses. These analyses consisted primarily of analyses of variance (one and two factor). Antimicrobial and antioxidant data were subjected to a two-factor ANOVA analysis of variance. Biochemical parameters were subjected to a one-factor ANOVA (group type) and blood glucose levels and induction, blood glucose treatment and body weight were subjected to a two-factor ANOVA (group type and measurement time). The Student–Newman–Keuls test was used for the separation of means considering a probability level of 5%. GraphPad Prism 8 software was used to create the graphs.

## 5. Conclusions

Plain or sprouted mung beans contains a variety of pharmacological properties, including antimicrobial, antioxidant and antidiabetic activities attributable to their nutrients and bioactive compounds. Phytochemicals in plain and sprouted seeds such as flavonoids and polyphenols have the ability to combat oxidative stress in the body by helping to maintain a balance between oxidants and antioxidants. This balance promoted by a proper diet is crucial in preventing chronic diseases such as diabetes that set in with oxidative damage. According to our results, mung beans grown in Burkina Faso, in their natural or sprouted state, lowered blood glucose levels in a dose-dependent manner in diabetic rats. The extracts also contributed to mitigate complications in diabetic rats by improving their lipid profile and significantly lowering urea levels, with the results even more pronounced with sprouted mung beans. In addition to its high nutritional value that can be used to prevent malnutrition in all its forms in Burkina Faso, the Beng-tigré variety of mung bean grown in Burkina Faso has multiple biological activities that meet the concept of a functional food that can contribute to the prevention and management of chronic diseases by reducing metabolic syndromes.

## Figures and Tables

**Figure 1 plants-11-03556-f001:**
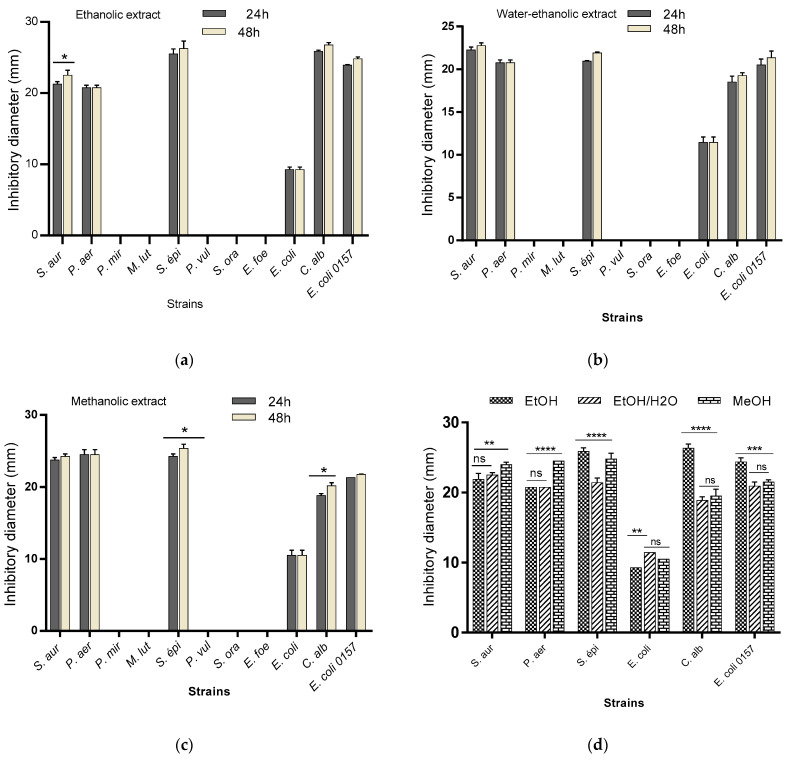
Inhibition capacity of ethanolic (**a**), hydroethanolic (**b**) and methanolic (**c**) extracts over time (24 h and 48 h) and the comparative effect of the extracts (**d**). *S. aur: Staphylococcus aureus; P. aer: Pseudomonas aeruginosa; P. mir: Proteus mirabilus; M. lut: Micrococcus luteus; S. epi: Staphylococcus epidermidis; P. vul: Proteus vulgaris; S. ora: Streptococcus oralis; E. fae: Enterococcus faecalis; E. coli: Escherichia coli ATCC25922; C. alb: Candida albicans; E. coli O157: Escherichia coli O157*. EtOH; ethanol extract; MeOH: methanol extract; EtOH/H_2_ O: hydroethanol extract. * *p* < 0.05; ** *p* < 0.01; *** *p* < 0.001; **** *p* < 0.0001.

**Figure 2 plants-11-03556-f002:**
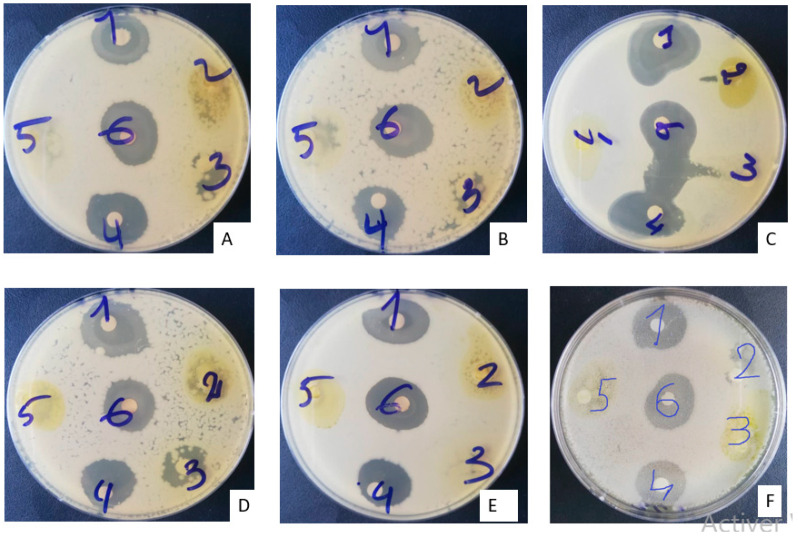
Appearance of antibiogram boxes of MBG extracts after 24 h of incubation. (**A**) *Escherichia coli* 0157 H7ATCC; (**B**) *Escherichia coli* ATCC 25922; (**C**) *Staphylococcus aureus* ATCC 29213; (**D**) *Candida albicans* MHMR; (**E**) *Staphylococcus epidermidis* T22695; (**F**) *Pseudomonas aeruginosa* ATCC 27853.

**Figure 3 plants-11-03556-f003:**
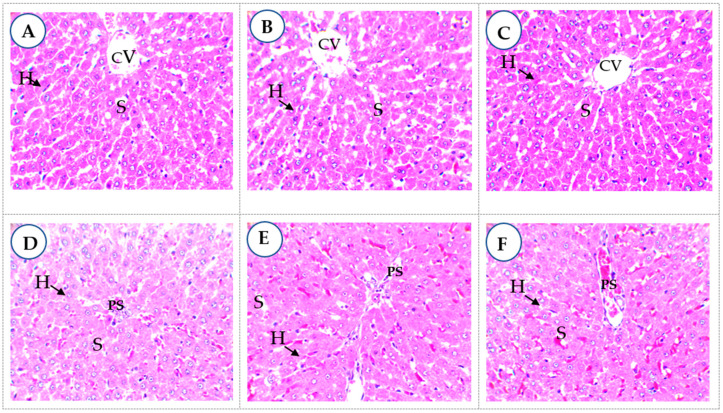
Liver parenchyma in subchronic oral toxicity test (400× magnification). CV: centrilobular veins; S: sinusoids; PS: portal spaces; H: hepatocytes. (**A**): non-diabetic control; (**B**): MBG extract at 150 mg/kg.bw; (**C**): MBN extract at 150 mg/kg.bw; (**D**): metformin at 100 mg/kg bw; (**E**): MBG extract at 300 mg/kg.bw; (**F**): MBN extract at 300 mg/kg.bw.

**Table 1 plants-11-03556-t001:** Phytochemical screening of MBN and MBG.

Compound Group	Class	MBN	MBG
**Polyphenolic compounds**	Tannins	**+**	**−**
	Catechic tannins	**+**	**−**
	Gallic tannins	**+**	**−**
	Flavonoids	**+**	**+**
	Anthocyanins	**+**	**+**
	Cyanogen derivatives	**−**	**−**
	Quinonic derivatives	**−**	**−**
**Heterosides**	Saponosides	**+**	**+**
	Reducing compound	**−**	**+**
	Mucilage	**+**	**+**
	O-heterosides	**−**	**−**
**Terpene compounds**	Terpenoids	**+**	**+**

+: presence; −: absence.

**Table 2 plants-11-03556-t002:** Minimum Inhibitory Concentrations (MICs) and Minimum Bactericidal Concentrations (MBCs) of extracts.

Extracts	Parameters (mg/mL)	Strain					
		*S. aur*	*P. aer*	*S. epi*	*E. coli*	*C. alb*	*E. coli O157*
Ethanolic	CMI	25	12.5	3.12	12.5	12.5	25
	CMB	50	>100	6.25	100	25	100
	CMB/CMI	2 *	-	2 *	8	2 *	4
Hydroethanolic	CMI	12.5	6.25	12.5	25	12.5	50
	CMB	25	>100	12.5	>100	25	100
	CMB/CMI	2 *	-	1 *	-	2 *	2 *
Methanolic	CMI	25	12.5	12.5	3.12	12.5	25
	CMB	50	>100	12.5	>100	25	100
	CMB/CMI	2 *	-	1 *	-	2 *	4

MIC: minimum inhibitory concentration, MBC: minimum bactericidal concentration; with *: bactericidal/fungicidal effect; without *: bacteriostatic/fungistatic effect. *S. aur: Staphylococcus aureus; P. aer: Pseudomonas aeruginosa; P. mir: Proteus mirabilus; M. lut: Micrococcus luteus; S. épi: Staphylococcus epidermidis; P. vul: proteus vulgaris; S. ora: Streptococcus oralis; E. fae: Enterococcus faecalis; E. coli: Escherichia coli ATCC25922; C. alb: Candida albicans; E. coli O157: Escherichia coli O157.*

**Table 3 plants-11-03556-t003:** Antioxidant activity of extracts and reference molecules.

	Extracts	DPPH			FRAP	
		%Inh	IC_50_ (mg/mL)	AAI	%Inh	IC_50_ (mg/mL)
**MBN**	Ethanolic	nd	-	-	77.51 ± 0.71	26.33 ± 0.57
	Water–ethanol	85.70 ± 0.06	28.65 ± 0.21	1.74 ± 0.01	81.9 ± 0.01	34.66 ± 0.57
	Methanolic	61.80 ± 0.19	78.50 ± 2.12	0.63 ± 0.01	83.40 ± 0.13	34.41 ± 0.52
**MBG**	Ethanolic	54.26 ± 0.06	80.00 ± 4.24	0.62 ± 0.03	82.25 ± 0.31	22.66 ± 2.08
	Water–ethanol	88.32 ± 0.06	32.50 ± 0.70	1.53 ± 0.03	82.50 ± 0.47	11.06 ± 0.40
	Methanolic	59.41 ± 0.89	38.00 ± 0.00	1.31 ± 0.00	81.06 ± 0.45	43.85 ± 0.49
**Reference**	BHA	91.28 ± 0.05	2.18 ± 0.47	23.43 ± 5.08	nd	nd
	Quercetin	nd	nd	nd	82.29 ± 0.25	1.98 ± 0.18

MBN: mung bean nature; MBG: mung bean germinated; BHA: Buthyhydroxyanisol; nd: not determined; AAI: antioxidant activity index.

**Table 4 plants-11-03556-t004:** Effect of ethanolic extract of mung bean on fasting blood glucose and body weight in rats made diabetic by STZ.

		G1	G2	G3	G4	G5	G6	G7
**Blood glucose treatment**	**D1**	113.50 ± 2.59 a	408.50 ± 6.63 ab	423.33 ± 44.32 a	406.66 ± 53.61 a	459.00 ± 18.17 a	434.33 ± 22.67 a	425.00 ± 24.82 a
	**D7**	105.00 ± 3.46 a	393.50 ± 13.56 b	399.66 ± 19.12 a	283.33 ± 94.69 a	259.66 ± 84.61 ab	167.33 ± 39.71 b	267.00 ± 19.05 b
	**D14**	104.00 ± 1.15 a	422.00 ± 2.30 a	270.33 ± 71.39 a	177.00 ± 93.04 a	113.00 ± 11.06 b	108.00 ± 2.64 b	166.50 ± 20.49 c
	**Average**	107 ± 1.98 D	408.00 ± 60.03 A	364.44 ± 34.41 BA	289.00 ± 53.00 BC	277.22 ± 56.10 BC	236.55 ± 51.89 C	286.16 ± 39.13 BC
	** *F-value* **	2.72	6.30	2.99	3.60	11.66	48.65	29.04
	**Probability**	*p* > 0.05	*p* < 0.05	*p* > 0.05	*p* > 0.05	*p* < 0.05	*p =* 0.001	*p* < 0.001
**Body weight**	**D1**	199.00 ± 8.66 c	200.00 ± 4.61 a	199.00 ± 5.29 a	199.33 ± 14.84 b	200.33 ± 6.35 b	199.66 ± 6.88 b	199.50 ± 8.37 a
	**D7**	201.00 ± 8.08 b	199.00 ± 4.61 b	198.33 ± 6.17 a	199.66 ± 14.74 b	201.66 ± 6.11 b	204.00 ± 4.50 ab	202.00 ± 8.66 b
	**D14**	204.00 ± 8.66 a	196.50 ± 4.90 c	200.66 ± 5.92 a	203.33 ± 14.51 a	205.33 ± 6.04 a	207.33 ± 5.69 b	205.50 ± 8.37 c
	**Average**	201.33 ± 4.29 A	198.50 ± 2.41 A	199.33 ± 2.92 A	200.77 ± 7.38 A	202.44 ± 3.18 A	203.66 ± 3.09 A	202.33 ± 4.32 A
	** *F-value* **	57.00	117.00	6.50	11.57	18.10	10.23	327.00
	**Probability**	*p* < 0.001	*p* < 0.0001	*p* > 0.05	*p* < 0.05	*p* < 0.01	*p* < 0.05	*p* < 0.001

G1: non-diabetic control, G2: diabetic control, G3: MBG 150 mg/kg.bw; G4: MBG 300 mg/kg.bw; G5: MBN 150 mg/kg.bw; G6: MBN 300 mg/kg.bw; G7: metformin 100 mg/kg.bw; D1: 1st day of follow-up; D7: 7th day of follow-up; D14: 14th day of follow-up. Different lowercase letters (a, b and c) between rows and columns indicate a significant difference. Different capital letters (A, B, C and D) between the rows of average for each parameter (blood glucose treatment and body weight) indicate a significant difference. The results are the average of 4 assays with 4 rats from each group (blood glucose treatment) and weight (body weight) of 4 rats from each group.

**Table 5 plants-11-03556-t005:** Variation in Erythrocytometry and Leukocytometry.

Parameters	Groups								
	G1	G2	G3	G4	G5	G6	G7	*F-value*	Probability
**HB** (g/dL)	19.00 ± 0.00 a	17.15 ± 0.08 a	18.63 ± 0.77 a	17.83 ± 0.78 a	17.63 ± 0.66 a	15.86 ± 1.23 a	18.3 ± 0.98 a	1.69	*p* > 0.05
**HT** (%)	58.00 ± 0.00 a	52.5 ± 0.28 a	57.00 ± 2.30 a	54.66 ± 2.33 a	54.00 ± 2.00 a	48.66 ± 3.71 a	56.00 ± 2.88 a	1.72	*p* > 0.05
**RBC** (g/L)	6.38 ± 0.00 a	5.77 ± 0.03 a	6.24 ± 0.28 a	6.01 ± 0.25 a	5.94 ± 0.22 a	5.33 ± 0.38 a	6.16 ± 0.31 a	1.75	*p* > 0.05
**WBC** (g/L)	5.8 ± 0.76 a	7.25 ± 0.95 a	6.66 ± 0.83 a	6.89 ± 0.94 a	6.62 ± 1.16 a	7.44 ± 1.12 a	6.18 ± 1.00 a	0.32	*p* > 0.05
**L** (%)	30.00 ± 2.82 a	24.00 ± 5.65 b	20.66 ± 4.16 b	27.33 ± 12.22 a	18.00 ± 2.00 b	20.00 ± 8.66 b	10.00 ± 2.82 c	3.12	*p* < 0.05
**N** (%)	70.00 ± 2.82 d	76.00 ± 5.65 c	79.33 ± 4.16 b	73.00 ± 12.76 d	81,66 ± 2.51 b	79.66 ± 8.96 b	90.00 ± 2.82 a	2.75	*p* < 0.05
**E** (%)	00.00 ± 0.00 a	00.00 ± 0.00 a	00.00 ± 0.00 a	00.00 ± 0.00 a	00.00 ± 0.00 a	00.00 ± 0.00 a	00.00 ± 0.00 a	0.18	*p* > 0.05
**M** (%)	00.00 ± 0.00 a	00.00 ± 0.00 a	00.00 ± 0.00 a	00.00 ± 0.00 a	00.00 ± 0.00 a	0.33 ± 0.57 a	00.00 ± 0.00 a	0.16	*p* > 0.05

G1: non-diabetic control; G2: diabetic control; G3: MBG 150 mg/kg.bw; G4: MBG 300 mg/kg.bw; G5: MBN 150 mg/kg.bw; G6: MBN 300 mg/kg.bw; G7: metformin 100 mg/kg.bw; HB: hemoglobin; HT: hematocrits; RBC: red blood cells; WBC: white blood cells; L: leukocytes; N: neutrophils; E: eosinophils; M: monocytes. The results are the average of 4 assays with 4 rats from each group. The probability *p* > 0.05 indicates that there is no variation between the groups for a dosed parameter (comparison on the line). In contrast, the probability *p* < 0.05 indicates a variation in the dosed parameter between the groups. Different lowercase letters (a, b, c and d) between rows and columns indicate a significant difference.

**Table 6 plants-11-03556-t006:** Variation in lipid profile of rats.

	Lipid Parameters Sought		
	Total Cholesterol (mmol/L)	Total Triglyceride (mmol/L)	LDL (mmol/L)
**G1**	3.41 ± 0.05 e	3.74 ± 0.01 d	0.20 ± 0.01 d
**G2**	4.37 ± 0.05 a	4.80 ± 0.01 a	0.51 ± 0.01 a
**G3**	4.08 ± 0.01 b	4.56 ± 0.02 b	0.32 ± 0.01 b
**G4**	3.90 ± 0.02 c	4.29 ± 0.02 c	0.27 ± 0.01 c
**G5**	3.53 ± 0.01 d	3.67 ± 0.02 d	0.26 ± 0.01 c
**G6**	3.36 ± 0.02 e	3.58 ± 0.01 e	0.23 ± 0.01 cd
**G7**	3.52 ± 0.01 d	3.72 ± 0.04 d	0.25 ± 0.00 c
** *F-value* **	177.98	430.93	88.57
**Probability**	*p* < 0.001	*p* < 0.001	*p* < 0.001

G1: non-diabetic control, G2: diabetic control, G3: MBG 150 mg/kg.bw; G4: MBG 300 mg/kg.bw; G5: MBN 150 mg/kg.bw; G6: MBN 300 mg/kg.bw; G7: metformin 100 mg/kg.bw; Different lowercase letters (a, b, c, d and e) between rows and columns indicate a significant difference.

**Table 7 plants-11-03556-t007:** Variation in transaminases (AST and ALAT) and urea and creatine levels.

	Hepatic Parameters		Renal Parameters	
	ASAT (U.I/L)	ALAT (U.I/L)	Urea (g/L)	Creatine (mg/L)
**G1**	109 ± 4.04 a	86 ± 7.50 a	0.25 ± 0.06 b	8.40 ± 1.84 b
**G2**	123 ± 0.00 a	91 ± 5.19 a	0.52 ± 0.00 a	16.55 ± 0.20 a
**G3**	96.66 ± 4.33 ba	88.33 ± 6.93 a	0.44 ± 0.06 ba	13.36 ± 1.70 ba
**G4**	70.33 ± 12.91 b	64.33 ± 13.48 a	0.29 ± 0.03 ba	9.2 ± 0.90 ba
**G5**	105.66 ± 3.48 a	89.66 ± 8.41 a	0.35 ± 0.09 ba	10.86 ± 2.85 ba
**G6**	72.66 ± 9.52 b	67.66 ± 10.72 a	0.34 ± 0.02 ba	13.7 ± 0.81 ba
**G7**	99 ± 1.73 ba	85 ± 2.30 a	0.36 ± 0.02 ba	11.85 ± 1.01 ba
** *F-value* **	7.41	1.55	2.90	3.21
**Probability**	*p =* 0.001	*p* > 0.05	*p* > 0.05	*p* < 0.05

*G1: non-diabetic control; G2: diabetic control; G3: MBG 150 mg/kg.bw; G4: MBG 300 mg/kg.bw; G5: MBN 150 mg/kg.bw; G6: MBN 300 mg/kg.bw; G7: Metformin 100 mg/kg.bw; ASAT: aspartate amino transferase; ALAT: alanine amino transferase.* Different lowercase letters (a, b) between rows and columns indicate a significant difference.

**Table 8 plants-11-03556-t008:** Pearson’s linear correlation.

	TChol	TryG	LDL	HB	HT	RBC	WBC	Urea	Créa	ASAT
**TryG**	0.977 0.000 ***									
**LDL**	0.868 0.000 ***	0.821 0.000 ***								
**HB**	0.033 0.886 ns	0.058 0.801 ns	−0.185 0.422 ns							
**HTE**	0.036 0.877 ns	0.061 0.794 ns	−0.186 0.420 ns	1.000 0.000 ***						
**NR**	0.036 0.877 ns	0.057 0.806 ns	−0.185 0.422 ns	0.999 0.000 ***	0.999 0.000 ***					
**NB**	0.183 0.427 ns	0.115 0.618 ns	0.147 0.525 ns	0.033 0.889 ns	0.033 0.886 ns	0.031 0.893 ns				
**Urea**	0.564 0.008 **	0.490 0.024 *	0.637 0.00 2 **	−0.082 0.725 ns	−0.081 0.726 ns	−0.077 0.741 ns	0.148 0.521 ns			
**Creatine**	0.428 0.053 ns	0.363 0.106 ns	0.563 0.008 **	−0.205 0.374 ns	−0.207 0.368 ns	−0.204 0.376 ns	0.059 0.801 ns	0.926 0.000 ***		
**ASAT**	0.291 0.201 ns	0.261 0.254 ns	0.473 0.030 *	0.260 0.254 ns	0.256 0.263 ns	0.268 0.240 ns	−0.192 0.405 ns	0.315 0.165 ns	0.219 0.340 ns	
**ALAT**	0.132 0.570 ns	0.145 0.532 ns	0.245 0.285 ns	0.320 0.158 ns	0.318 0.160 ns	0.325 0.151 ns	−0.124 0.593 ns	0.209 0.364 ns	0.106 0.648 ns	0.844 0.000 ***

*ns: not significant (p > 0.05); * p < 0.05; ** p < 0.01; *** p < 0.001. ASAT: aspartate amino transferase; ALAT: alanine amino transferase. HB: hemoglobin, HT: hematocrit, RBC: red blood cell, WBC: white blood cell.*

## Data Availability

Not applicable.

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
