# Peer review of "Mung Bean (Vigna radiata (L.) R. Wilczek) from Burkina Faso Used as Antidiabetic, Antioxidant and Antimicrobial Agent"

_plants, 2022, doi:10.3390/plants11243556_

Round 1

Reviewer 1 Report

Please provide Mung bean [Vigna radiata (L.)R. Wilczek} clinical research papers published in the SCI area of hypoglycemic and  diabetes treatment, please provide the effect of {Vigna radiata (L.) R .Wilczek ] on HbA1c.

Reviewer 2 Report

Dear authors, I suggest you make some corrections.

a) Numbers and units of measurement appear in the abstract, they must leave a space between the numbers and the units.

b) This error is repeated throughout the text on several occasions: 24 h; 48 hours; lines 305-309; 390 (60°C); 392 (60°C); 396 (of (31)); 435; 442-446.

c) In lines 89-90 you must put the full name of the genus of the bacterium or fungus the first time you cite it. You must also correct this between lines 110-116.

d) At the bottom of figure 1: Proteus (capital letters and italics); S. épid (without tilde).

e) Properly order the data in the last rows of table 3.

f) Kg is written with the letter k in lower case. Correct this unit on lines 171,173,177, 187,190,192,194.

g) In table 5 you must specify what it means: J1, J7, J14

h) The first time you use the abbreviation bw you should indicate what it means.

i) You must write the bibliography uniformly, for example: the last author preceded by and or by &, whichever is required by the journal. Take care of the use of commas (only after the surname or only between different authors...).

Yours faithfully.

Reviewer 3 Report

The authors in the article of " Mung bean [Vigna radiata (L.) R. Wilczek] from 2 Burkina-Faso used as antidiabetic, antioxydant and an- 3 timicrobial agent" tried to discuss different application of Mung Bean and in general did a good work. However, the following topics need to address: 

1. Introduction is too short with almost no discussion about the background of the work. Tones of articles already published about the Mung Bean  and it is necessary to address what is new in this work.

2. The most important part of this type of work if detection and content of extracted chemicals from the plant and correlation of investigated properties with extracted chemicals. there ris no quantitative number for content (mg/g) of extracted chemicals using different solvents to correlate with the results. 

3. Despite good work about antidiabetic, antioxydant and an- 3 timicrobial agent, it is not clear how these properties correlated with type of chemicals in the bean. Therefore, it is hard to make a general conclusion with wide range of application. 

Round 2

Reviewer 3 Report

1. As I mentioned in my first review, the seed is investigated by different researchers with good details of extracted chemicals. Why there is no information about them in the introduction section to compare with your work results. 

2. due to no extracted chemical content provided by authors, it is necessary to mention chemical composition of the seed reported by other researchers in the introduction section. Without this data, it is hard to make a conclusion about the reported properties.  
